# The Microstructural Model of the Ferromagnetic Material Behavior in an External Magnetic Field

**Anatoli A. Rogovoy** *[ID], **Oleg V. Stolbov** [ID] and **Olga S. Stolbova** [ID]

Institute of Continuous Media Mechanics of the Ural Branch of Russian Academy of Sciences,
614018 Perm, Russia; sov@icmm.ru (O.V.S.); sos@icmm.ru (O.S.S.)
* Correspondence: rogovoy@icmm.ru

**Abstract:** In this paper, the behavior of a ferromagnetic material is considered in the framework of microstructural modeling. The equations describing the behavior of such material in the magnetic field, are constructed based on minimization of total magnetic energy with account of limitations imposed on the spontaneous magnetization vector and scalar magnetic potential. This conditional extremum problem is reduced to the unconditional extremum problem using the Lagrange multiplier. A variational (weak) formulation is written down and linearization of the obtained equations is carried out. Based on the derived relations a solution of a two-dimensional problem of magnetization of a unit cell (a grain of a polycrystal or a single crystal of a ferromagnetic material) is developed using the finite element method. The appearance of domain walls is demonstrated, their thickness is determined, and the history of their movement and collision is described. The graphs of distributions of the magnetization vector in domains and in domain walls in the external magnetic field directed at different angles to the anisotropy axis are constructed and the magnetization curves for a macrospecimen are plotted. The results obtained in the present paper (the thickness of the domain wall, the formation of a 360-degree wall) are in agreement with the ones available in the current literature.

**Keywords:** ferromagnetic material; micromagnetism; variational formulation; finite element method





## 1. Introduction

The ferromagnetic Heusler alloy $Ni_2MnGa$ is of great interest due to its unique ability to produce significant deformation (up to 6–10%) in the martensitic (low-temperature) state under the action of a moderate magnetic field [1]. This deformation does not disappear when the magnetic field is removed, but becomes reversible as the material is being brought back to the austenitic (high-temperature) state (the effect of shape memory). Thus, there exists a possibility of controlling the deformation behavior of the material using a magnetic field. Moreover, the response of the material to the applied magnetic field is almost instantaneous, which makes this control inertialess. In recent year, the $Ni_2MnGa$ alloy has been the focus of extensive research aimed at designing new functional materials, which change their shape and size under the action of an external magnetic field and restore them as a result of reverse phase transition, which can also be controlled by the magnetic field, yet of much higher magnitude.

If in the shape memory alloys (SMAs) or ferromagnetic shape memory alloys (FSMAs) the forward first-order phase transition from the austenitic to the martensitic state (cooling) is realized only due to temperature changes, in the martensitic state there may exist several variants of martensite. Pairs of martensite variants form twinned structures, which are so compatible that can experience only volume deformation of insignificant value. The application of stress or an external magnetic field in this state causes traditional elastic deformation in the SMA and insignificant magnetostriction deformation in the FSMA, corresponding to the applied forces and magnetic field. Occurrence of these traditional deformations is complemented by the process of structure detwinning, which leads to the

development of significant deformation, 6–10 times higher than the elastic one. In the case when in SMA or in FSMAs the forward phase transition proceeds under uniaxial tension caused by the applied constant external force, in the martensitic state there is only one variant of martensite (structure detwinning during phase transition) consistent with the stress field in the specimen. Deformation developed in the single martensite variant is the ordinary elastic deformation corresponding to the force applied to the material and supplemented by phase deformation, which, as in the above case, is 6–10 times higher than the elastic one. Since in this case detwinning of martensite structures accompanied by large deformations takes place during phase transition, the application of stress or an external magnetic field in the martensitic state cause only ordinary elastic deformation or magnetostriction deformation.

In modern mechanics of a deformable solid, one can distinguish two fundamental problems, which are directly associated with practical applications. The first problem is the development of an approach to the construction of correct equations for treatment of thermo-elastic-inelastic deformation behavior of complex media at finite deformations. The correct equations are those equations that satisfy the principles of thermodynamics and objectivity (material independence from the choice of reference system). Today, the construction of such equations is rather the mastery. Therefore, it would be proper to develop rules, i.e., a kind of operation algorithm, the result of which would be a correctly formulated model, describing the behavior of the examined medium. The second problem is the construction of models to describe the behavior of material based on its structure and structural changes caused by force and/or thermal, and/or magnetic and other external actions (physical or micromechanical models, or structural and analytical models according to Likhachev's nomenclature). Such models are required for adequate interpretation of the behavior of materials, in which phase or structural transitions occur under the above mentioned external actions, in particular, for shape memory alloys, including ferromagnetic shape memory alloys (of Heusler type). Solution of the first problem allows one to correctly describe the kinematics and construct the governing equations for each structural element of such models (grains, which are single crystals in a polycrystalline material). Physical (micromechanical) models, in contrast to the widespread phenomenological models, allow us to explicitly consider the physical processes occurring in the structural elements of the material. It makes possible to dispense with many of the hypotheses used to construct phenomenological models, and to give a physically meaningful description of the behavior of complex functional materials undergoing finite deformations and structural changes under the action of temperature, force and magnetic fields.

One of the variants of solving the first problem is proposed in [2]. It is called a formalized approach to the construction of constitutive equations and describes the behavior of complex media with finite deformations and structural changes in the material. This approach was used in [3] for modeling the process of controlling the temperatures of phase transition in FSMAs using a magnetic field. This field shifts these temperatures according to the generalized Clausius-Clapeyron law, the form of which for the problems of mechanics was also formulated in the above publication. In this study, the forward phase transition was realized under the action of a constant uniaxial tensile force, which was removed in the martensitic state. Such a process, as noted above, is not accompanied by the formation of twins. So, the application of an external magnetic field in the martensitic state leads only to a slight magnetostrictive deformation of the crystal cell. The reorientation of martensite variants and their detwinning by means of a magnetic field, inducing significant deformations (the main factor determining practical interest to these alloys), does not take place in this process. Therefore, a natural continuation of work [3] is consideration of the influence of the magnetic field on the reorientation of the martensite. The first thing to do in this regard and the main objective of the present study is the construction of a model to describe the behavior of polycrystalline material in an external magnetic field, taking into account the motion of the boundaries of magnetic domains and the rotation of the magnetic vector moment in each mesoelement, representing a grain of material. The above stated

problems can be solved only within the framework of the microstructural approach [4,5], which, in contrast to the widely widespread phenomenological models [6,7], allows us to explicitly take into account the physical processes occurring in structural elements of the material without recourse to the assumptions commonly used to construct phenomenological models. In mechanics of deformable solids, which is the major focus of our research, the magnetization vector in the existing microstructural models of thermo-magneto-elastic behavior of shape memory alloys abruptly changes at the boundary of magnetic domains. These models ignore the fact that the domain wall has a certain thickness and the magnetization vector continuously undergoes changes through thickness (see, for example, works [8–10]). With that knowledge in mind, it is possible to refine the known microstructural models and substantiate the need for such refinement. This is what we plan to do in the framework of deformable solid mechanics. In this respect, the present study is a necessary step in this direction aimed to construct within only the framework of the theory of magnetism a microstructural model, in which one of the structural elements is a domain wall of a certain thickness. Initially, the domain wall in the sample has a zero thickness. Minimizing the functional of magnetic energy in the absence of an external magnetic field, we construct a domain wall of finite thickness and obtain the distribution of the magnetization vector in it. Taking such a domain structure, we study its changes (motion of domain walls and their interaction) under the action of an external magnetic field applied in different directions.

This work is an integral part of a more general study, the ultimate goal of which is to describe the behavior of shape memory alloys both in terms of their magnetic properties and mechanical ones. For this reason, we considered it necessary to give above a detailed description of the whole problem to demonstrate the place, which the research, presented in this paper, occupies in the entire problem. However, the approach developing in this paper, which is based on minimizing the magnetic energy functional, is applicable not only to shape memory alloys, but also to other magnetic materials.

## 2. Structure of a Shape-Memory Material Crystal Cell, Magnetic Domains

In the shape memory alloys such as *NiTi* alloy, in which the austenite crystal has a body-centered cubic structure, the process of forward phase transition induced solely by temperature variation (cooling) can give rise to about 12 variants of martensite with a base-centered monoclinic for some alloys crystal structure of different spatial orientation (Monoclinic crystal cell is an inclined parallelepiped with different edges $a$, $b$ and $c$ and a rectangular base). This follows from the fact that the symmetry group (parity) of a cubic lattice consists of 24 orthogonal tensors, which transform a cube into the same cube, and the symmetry group of a monoclinic lattice, which is a subgroup of the first group, consists of two orthogonal tensors transforming the inclined parallelepiped with different edges and rectangular base into the same parallelepiped. The transformation of a cubic lattice to a monoclinic one corresponds to the Bain strain tensor. Since there are 24 equivalent cubic lattices, the number of different Bain tensors (different variants of martensite) can also amounts to 24, but only 12 of them will be independent due to the indiscernibility of two configurations of the monoclinic lattice. Each of the 12 variants of martensite has the same Bain tensor referred to its own crystallographic axes, which are differently oriented in the space.

In contrast to the *NiTi* shape memory alloy, in the $Ni_2MnGa$ FSMA, the cubic austenite crystal is transformed during forward temperature-driven phase transition (without a stress field) only into three variants of martensite with a tetragonal crystal cell, in which $a$, $a$ and $c$, $a > c$ are the cell edges, with the short edge being directed parallel to the three edges of the austenitic cube. Since the martensitic state temperature is much lower than the Curie temperatures, the martensite variants are spontaneously magnetized even in the absence of an external magnetic field. The local magnetization vector (magnetic moment) in each variant of martensite is oriented along one of the crystallographic directions, called the axis of easy magnetization, and can be directed both along this axis and against it. For a

tetragonal crystal cell, this axis coincides with the short edge $c$. The regions composed of the interconnected martensite variants, in which the magnetization vectors are oriented in one direction, form magnetic domains. As a result, many magnetic domains with differently directed magnetization vectors are formed in the martensitic state. For these domains it is energetically favorable to be so compatible with each other that the total magnetization of the material is zero (for more information about domains, see [11]). The application of an external magnetic field set in motion the walls of magnetic domains, causes rotation of the magnetization vectors and reorientation of martensite variants. The wall of the magnetic domain has certain thickness. In this region the magnetization vector of one domain gradually rotates until its direction coincides with the direction of the magnetization vector of adjacent domain. Under the action of the applied magnetic field the domain, in which the magnetization vector is more consistent with the field vector, begins to grow at the expense of the adjacent domain, in which the magnetization vector is less consistent with the direction of the applied field, i.e., the domain wall begins to move. When this mechanism becomes exhausted, the magnetization vectors in the domains are set in rotational motion in the direction of the applied magnetic field. These two mechanisms are inherent in conventional ferromagnetic materials. They cause a certain, usually insignificant, macrodeformation (magnetostriction), which disappears when the external magnetic field is removed. The reorientation of martensite variants under the action of the magnetic field is inherent only in FSMA and, in fact, is similar to reorientation and detwinning, occurring in the SMA under the action of a force field. A cooperative movement of structural elements of the material causes significant macrodeformations (6–10%), which do not disappear after removing the magnetic field. In the event of realization of reverse phase transition (heating of the material and its transition to the austenitic state), all accumulated strains are cancelled out.

In the following sections, we model the behavior of such a material in an external magnetic field using a microstructural approach. The need to use this approach is justified in the introduction.

## 3. Magnetic Energy

According to [12], at a temperature much lower than the Curie temperature of the material under consideration, the magnetization is described in terms of the time and space dependent vector field of spontaneous magnetization **M**, such that $|\mathbf{M}| = M_s$, where $M_s$ is the saturation magnetization. It is necessary to find the spatial distribution of the magnetization vector **M** in the equilibrium state.

Let us introduce a unit vector $\mathbf{m} = \mathbf{M}/M_s$. Equilibrium in the "body—environment" system is found by minimizing the magnetic energy, which can be represented as the sum of four addends (see, for example, [8]):

$$\psi(\mathbf{p}, \mathbf{m}) = \psi_{ext}(\mathbf{m}) + \psi_{demag}(\mathbf{m}) + \psi_{exch}(\mathbf{m}) + \psi_{anis}(\mathbf{p}, \mathbf{m}), \tag{1}$$

where $\psi_{ext}(\mathbf{m})$ is the Zeeman energy, $\psi_{demag}(\mathbf{m})$ is the demagnetization energy, $\psi_{exch}(\mathbf{m})$ is the exchange energy, $\psi_{anis}(\mathbf{p}, \mathbf{m})$ is the magnetocrystalline anisotropy energy, **p** is the anisotropy axis.

The Zeeman energy is a potential energy of the magnetized body in the external magnetic field:

$$\psi_{ext}(\mathbf{m}) = -\mu_0 M_s \, \mathbf{H}_0 \cdot \mathbf{m},$$

where $\mu_0$ is the magnetic constant, $\mathbf{H}_0$ is the strength of the external magnetic field. The Zeeman energy causes the magnetization vector to be aligned with applied magnetic field.

The energy of demagnetization takes into account the interactions of all local magnetic moments in the system and is determined by the following relation:

$$\psi_{demag}(\mathbf{m}) = -\frac{1}{2}\mu_0 M_s \, \mathbf{H}_{demag} \cdot \mathbf{m}.$$

Here $\mathbf{H}_{demag}$ is the demagnetizing field strength. As a result, the strength of the operating field is $\mathbf{H} = \mathbf{H}_0 + \mathbf{H}_{demag}$. This field in the absence of electric currents must be vortex-free $\nabla \times \mathbf{H} = \mathbf{0}$, which for constant $\mathbf{H}_0$ reduces to the equality $\nabla \times \mathbf{H}_{demag} = \mathbf{0}$, which will always be true, if we assume that $\mathbf{H}_{demag} = -\nabla \varphi$, where $\varphi = \varphi(\mathbf{x})$ is the scalar dependent on the vector coordinate $\mathbf{x}$ (the minus sign in front of the scalar potential $\varphi$ is used to show that the demagnetizing field reduces the magnetic field inside the specimen $\mathbf{H} = \mathbf{H}_0 - \nabla \varphi$). The magnetic field inductance $\mathbf{B}$, which must satisfy the equation $\nabla \cdot \mathbf{B} = 0$, is introduced by the relation $\mathbf{B}(\mathbf{x}) = \mu_0 \, (\mathbf{H} + M_s \, \mathbf{m})$, for $\mathbf{x} \in \Omega^{(in)}$, where $\Omega^{(in)}$ is the region occupied by the body, and $\mathbf{B}(\mathbf{x}) = \mu_0 \, \mathbf{H}$ for $\mathbf{x} \in \Omega^{(ex)}$, where $\Omega^{(ex)}$ is the region occupied by the surrounding medium. With account of the above representations for $\mathbf{H}$, $\mathbf{H}_{demag}$ and the constancy of $\mathbf{H}_0$, this yields for the function $\varphi$ both the Poisson equation

$$\nabla \cdot \nabla \varphi = M_s \, \nabla \cdot \mathbf{m} \quad \forall \mathbf{x} \in \Omega^{(in)},$$

where it is assumed that $M_s = const$, which holds true if only at the examined structural level the body is considered to be formed of one and the same material, and the Laplace equation

$$\nabla \cdot \nabla \varphi = 0 \quad \forall \mathbf{x} \in \Omega^{(ex)}.$$

The natural requirement, which the function $\varphi$ must obey, is

$$\varphi \to 0 \quad \text{at} \quad \mathbf{x} \to \infty.$$

On the surface $\Gamma$, which separates the body from the surrounding medium,

$$\varphi^{(in)}|_\Gamma = \varphi^{(ex)}|_\Gamma,$$

where index $(in)$ refers to the body and $(ex)$ – to its exterior domain. Other boundary conditions specified for the function $\varphi$ are related to the behavior of the vectors $\mathbf{H}$ and $\mathbf{B}$ on the surface $\Gamma$:

$$(\mathbf{H}^{(in)} - \mathbf{H}^{(ex)})|_\Gamma \cdot \boldsymbol{\tau} = 0, \quad (\mathbf{B}^{(in)} - \mathbf{B}^{(ex)})|_\Gamma \cdot \mathbf{n} = 0,$$

where $\boldsymbol{\tau}$ and $\mathbf{n}$ are the unit tangent and normal (from the body toward the surrounding space). From the first equality it follows that

$$(\nabla \varphi^{(in)} - \nabla \varphi^{(ex)})|_\Gamma \cdot \boldsymbol{\tau} = 0,$$

and from the second equality we get

$$(\nabla \varphi^{(in)} - \nabla \varphi^{(ex)})|_\Gamma \cdot \mathbf{n} = M_s \, \mathbf{m} \cdot \mathbf{n}. \tag{2}$$

The exchange energy is a phenomenological continual description of exchange interaction in quantum mechanics:

$$\psi_{exch}(\mathbf{m}) = A_{exch} \, |\nabla \mathbf{m}|^2 = A_{exch} \, ((\nabla m_x)^2 + (\nabla m_y)^2 + (\nabla m_z)^2),$$

where $A_{exch}$ is the exchange constant, $m_x, m_y, m_z$ are the components of $\mathbf{m}$. The exchange energy tends to contribute to configurations, in which magnetization changes slowly throughout the specimen. This energy is minimized when the magnetization is perfectly homogeneous.

Magnetocrystalline anisotropy takes into account the dependence of the local magnetization on the directions of the preferred magnetization. The easy axes of local martensitic variants are oriented in different directions, so this energy depends on the relative orientation of $\mathbf{m}$ and easy axis. We restrict our consideration to the uniaxial case, when there is

only one easy axis in a variant. For the $\alpha$ variant, the magnetocrystalline anisotropy energy is written as

$$\psi_{anis}^{\alpha}(\mathbf{p}^{\alpha}, \mathbf{m}) = K_{anis}\left(1 - (\mathbf{m} \cdot \mathbf{p}^{\alpha})^2\right),$$

where $K_{anis}$ is the anisotropy constant, $\mathbf{p}^{\alpha}$ is the direction of the easy axis of the $\alpha$ variant. The anisotropy energy tends to create the magnetic configurations, in which the magnetization is aligned along the easy axis.

Thus, the magnetic energy (1) is represented as follows:

$$\psi(\mathbf{m}) = -\mu_0 M_s\,\mathbf{H}_0 \cdot \mathbf{m} - \frac{1}{2}\mu_0 M_s\,\mathbf{H}_{demag} \cdot \mathbf{m} +$$
$$A_{exch}\left((\nabla m_x)^2 + (\nabla m_y)^2 + (\nabla m_z)^2\right) + K_{anis}\left(1 - (\mathbf{m} \cdot \mathbf{p}^{\alpha})^2\right). \quad (3)$$

## 4. Statement of the Problem

In 1907, Weiss, on the assumption of the existence of magnetic domains in ferromagnets, developed the domain theory of ferromagnetism. Landau and Lifshitz [13] suggested an idea that the domain structure of the material minimizes its Gibbs free energy. Brown put together the concepts previously developed by Weiss, Landau and Lifshitz, and created a unified continuous theory for ferro- and ferrimagnetic systems, which he designated micromagnetism [12]. In this paper, the study of the dynamics of magnetic inhomogeneities (domain boundaries, domains, magnetic vortices, etc.) is carried out within the framework of this general micromagnetic theory. In this approach, the analysis is performed at intermediate scale, which, on the one hand, is small enough to capture the details of the structure of transition regions between the domains, and, on the other hand, is sufficiently large to make use of a continuous magnetization vector, rather than individual atomic spins. The description of magnetization dynamics and the study of the structural evolution of domain walls, as well as their dynamic behavior is conventionally performed in the framework of the Stoner–Wohlfarth model or the nonlinear Landau–Lifshitz vector equation. The Stoner–Wohlfarth model is a widely used model for the magnetization of single-domain ferromagnets [14]. In this model, the magnetization does not vary within the ferromagnet and is represented by a vector $\mathbf{M}$, which rotates as the magnetic field $\mathbf{H}$ changes. The energy of the system is represented as the sum of two terms: the first term is the magnetic anisotropy and the second one is the energy of coupling with the applied field (the Zeeman energy). The nonlinear Landau–Lifshitz vector equation is based on the fundamental law of mechanics—the law of variation of the angular momentum [12,15]. Being a solution of this equation, a solitary 180-degree domain boundary (*N*-degree domain boundary is a transition layer between the adjacent domains *A* and *B* with opposite or coinciding directions of the magnetization vectors $\mathbf{m}_A$ and $\mathbf{m}_B$, in which the magnetic moment gradually changes its direction from $\mathbf{m}_A$ to $\mathbf{m}_B$. If, in this case, the rotation of the magnetization vector occurs in the plane coinciding with the plane of the wall, the boundary is called a Bloch boundary, if this occurs in the plane perpendicular to the plane of the wall, then the boundary is called a Neel boundary [15].) is a one-soliton formation called a kink or topological soliton. Two strongly interacting domain walls are called a two-soliton formation. Such formations include spatially localized magnetic inhomogeneities in the form of a dynamic, zero-degree domain wall and a dynamic 360-degree domain wall [15]. However, the Stoner–Wohlfarth model and the Landau–Lifshitz equation are not the only tool for describing the dynamics of magnetization and the evolution of the domain wall structure. This can also be done based on the condition of the extremum of the total magnetic energy [15] by performing a stepwise loading procedure. This approach is most convenient, bearing in mind that the final aim of our study is to describe the behavior of shape memory alloys not only in terms of their magnetic properties, but also mechanical ones. This generates a need for constructing a model of material behavior, which can readily incorporate the relations describing elastic and inelastic (phase) strains within the framework of finite deformations. Because of the specific character of the examined problem, the application of the standard micromagnetic packages (OOMMF, MuMax, Magpar,

nmag) for its solution has proved to be unjustified. The numerical simulations made in the context of this problem were implemented using the universal computing platform for solving partial differential equations. For this reason, the focus of the study that follows is the construction of the functional, its minimization, and the derivation of variational equations, which are solved numerically by the finite element method in Section 4.

Let us consider a ferromagnetic specimen occupying the region $\Omega^{(in)}$, in which it is necessary to determine the spatial distribution of unit magnetization vector **m** (only its direction) from the condition of functional minimum

$$\Psi(\mathbf{m}(t)) = \int_{\Omega^{(in)}} \psi(\mathbf{m}(t)) \, d\Omega^{(in)}, \tag{4}$$

where $\psi(\mathbf{m})$ is defined by relation (3) under the following constraints:

$$\mathbf{H}_{demag} = -\nabla\varphi;$$
$$\nabla \cdot \nabla\varphi = M_s \nabla \cdot \mathbf{m} \quad \forall \mathbf{x} \in \Omega^{(in)};$$
$$\nabla \cdot \nabla\varphi = 0 \quad \forall \mathbf{x} \in \Omega^{(ex)};$$
$$m_x^2 + m_y^2 + m_z^2 = 1.$$

The conditional extremum problem (3), (4) is reduced to a problem of an unconditional extremum using the method of Lagrange multipliers

$$\Psi_L(\mathbf{m}(t), \varphi, \lambda_1, \lambda_2, \lambda_3) = \int_{\Omega^{(in)}} [-\mu_0 \, M_s \, \mathbf{H}_0 \cdot \mathbf{m} + \frac{1}{2} \mu_0 \, M_s \, \nabla\varphi \cdot \mathbf{m} +$$
$$+ A_{exch} \left((\nabla m_x)^2 + (\nabla m_y)^2 + (\nabla m_z)^2\right) + K_{anis} \left(1 - (\mathbf{m} \cdot \mathbf{p}^\alpha)^2\right)] \, d\Omega^{(in)} +$$
$$+ \int_{\Omega^{(in)}} \lambda_1(\mathbf{x})(\nabla \cdot \nabla\varphi - M_s \nabla \cdot \mathbf{m}) \, d\Omega^{(in)} + \int_{\Omega^{(ex)}} \lambda_2(\mathbf{x}) \nabla \cdot \nabla\varphi \, d\Omega^{(ex)} +$$
$$+ \int_{\Omega^{(in)}} \lambda_3(\mathbf{x})(m_x^2 + m_y^2 + m_z^2 - 1) \, d\Omega^{(in)}.$$

Variation of this functional is carried out with respect to the quantities **m**, $\varphi$ and $\lambda_i$ ($i = 1, 2, 3$), which depend on time $t$ and spatial coordinate **x**. As a result, we have

$$\delta\Psi_L = \int_{\Omega^{(in)}} [(-\mu_0 \, M_s \, \mathbf{H}_0 + \frac{1}{2} \mu_0 \, M_s \, \nabla\varphi - 2 \, K_{anis} \, (\mathbf{m} \cdot \mathbf{p}^\alpha)\mathbf{p}^\alpha - \lambda_1(\mathbf{x}) M_s \nabla) \cdot \delta\mathbf{m} +$$
$$+ 2 \, A_{exch} \left((\nabla m_x) \cdot (\nabla \delta m_x) + (\nabla m_y) \cdot (\nabla \delta m_y) + (\nabla m_z) \cdot (\nabla \delta m_z)\right) +$$
$$+ 2 \, \lambda_3(\mathbf{x})(m_x \delta m_x + m_y \delta m_y + m_z \delta m_z)] \, d\Omega^{(in)} +$$
$$+ \int_{\Omega^{(in)}} (\frac{1}{2} \mu_0 \, M_s \, \mathbf{m} + \lambda_1(\mathbf{x})\nabla) \cdot \nabla \, \delta\varphi \, d\Omega^{(in)} + \int_{\Omega^{(ex)}} \lambda_2(\mathbf{x}) \nabla \cdot \nabla \delta\varphi \, d\Omega^{(ex)} +$$
$$+ \int_{\Omega^{(in)}} (\nabla \cdot \nabla\varphi - M_s \nabla \cdot \mathbf{m}) \, \delta\lambda_1(\mathbf{x}) \, d\Omega^{(in)} + \int_{\Omega^{(ex)}} \nabla \cdot \nabla\varphi \, \delta\lambda_2(\mathbf{x}) \, d\Omega^{(ex)} +$$
$$+ \int_{\Omega^{(in)}} (m_x^2 + m_y^2 + m_z^2 - 1) \, \delta\lambda_3(\mathbf{x}) \, d\Omega^{(in)} = 0. \tag{5}$$

Here, the fourth and fifth lines contain the second derivatives of $\delta\varphi$ and $\varphi$ with respect to coordinates, which in the numerical simulation requires that the approximation used for these quantities should be not lower than the quadratic one. Taking into account the readily proved equality

$$\nabla \cdot (\alpha \, \mathbf{b}) = \mathbf{b} \cdot (\nabla\alpha) + \alpha \, (\nabla \cdot \mathbf{b}), \tag{6}$$

where $\alpha$ and **b** are the arbitrary scalar and vector, this requirement can be significantly weakened by using the so-called weak formulation (Weak formulation does not imply that instead of differential relations we use equations written in the variational form after applying to them the Galerkin method. This means that the equation obtained as a result of this procedure can be transformed to a form that significantly reduces the

smoothness requirements, and differentiability of the sought solution in comparison with the differential formulation).

Bearing in mind the fact that the Lagrange multiplier $\lambda_1(\mathbf{x})$ acts in the domain $\Omega^{(in)}$, and the multiplier $\lambda_2(\mathbf{x})$ acts in the domain $\Omega^{(ex)}$, and these regions do not intersect ($\Omega^{(in)} \cap \Omega^{(ex)} = 0$), we can change the above situation, using instead of two Lagrange multipliers, a single one, which acts in the combined domain $\Omega^{(in)} \cup \Omega^{(ex)}$ and is continuous on the surface $\Gamma$ separating these domains. Then, the fourth line in (5) can be represented as

$$\int_{\Omega^{(in)}} (\frac{1}{2} \mu_0 M_s \mathbf{m} + \lambda(\mathbf{x}) \nabla) \cdot \nabla \delta\varphi \, d\Omega^{(in)} + \int_{\Omega^{(ex)}} \lambda(\mathbf{x}) \nabla \cdot \nabla \delta\varphi \, d\Omega^{(ex)},$$

which after applying the equality (6) is written as follows:

$$\int_{\Omega^{(in)}} (\frac{1}{2} \mu_0 M_s \mathbf{m} \cdot \nabla \delta\varphi + \nabla \cdot (\lambda(\mathbf{x}) \nabla \delta\varphi) - (\nabla \delta\varphi) \cdot (\nabla \lambda)) \, d\Omega^{(in)} +$$
$$+ \int_{\Omega^{(ex)}} (\nabla \cdot (\lambda(\mathbf{x}) \nabla \delta\varphi) - (\nabla \delta\varphi) \cdot (\nabla \lambda)) \, d\Omega^{(ex)}.$$

From this expression, in view of the Ostrogradsky–Gauss theorem, we get

$$\int_{\Gamma} \lambda(\mathbf{x})(\mathbf{n}^{(in)} \cdot \nabla \delta\varphi^{(in)}) \, d\Gamma + \int_{\Omega^{(in)}} (\frac{1}{2} \mu_0 M_s \mathbf{m} - \nabla \lambda) \cdot \nabla \delta\varphi \, d\Omega^{(in)} +$$
$$+ \int_{\Gamma \cup \Gamma_\infty} \lambda(\mathbf{x})(\mathbf{n}^{(ex)} \cdot \nabla \delta\varphi^{(ex)}) \, d\Gamma - \int_{\Omega^{(ex)}} (\nabla \delta\varphi) \cdot (\nabla \lambda) \, d\Omega^{(ex)}. \quad (7)$$

Here, $\mathbf{n}^{(in)}$ and $\mathbf{n}^{(ex)}$ are the unit normals to the surfaces bounding the domains $\Omega^{(in)}$ and $\Omega^{(ex)}$, respectively, which are directed outward, $\Gamma$ is the surface separating these regions. By imposing the condition $\varphi \to 0$ on $\Gamma_\infty$ on the function $\varphi$, taking into account the notation of the normal $\mathbf{n}^{(in)}$ ($\mathbf{n}^{(in)} = \mathbf{n}$) and the relationship between $\mathbf{n}^{(in)}$ and $\mathbf{n}^{(ex)}$ on $\Gamma$, we can transform the expression (7) to

$$\int_{\Gamma} \lambda(\mathbf{x}) \mathbf{n} \cdot (\nabla \delta\varphi^{(in)} - \nabla \delta\varphi^{(ex)}) \, d\Gamma + \int_{\Omega^{(in)}} \frac{1}{2} \mu_0 M_s \mathbf{m} \cdot \nabla \delta\varphi \, d\Omega^{(in)} -$$
$$- \int_{\Omega^{(in)} \cup \Omega^{(ex)}} (\nabla \delta\varphi) \cdot (\nabla \lambda) \, d\Omega, \quad (8)$$

in which the integrals over the common surface $\Gamma$ are grouped together. Finally, with account of equality (2), the expressions (8) take the following form

$$\int_{\Gamma} \lambda(\mathbf{x}) \mathbf{n} \cdot M_s \, \delta\mathbf{m} \, d\Gamma + \int_{\Omega^{(in)}} \frac{1}{2} \mu_0 M_s \mathbf{m} \cdot \nabla \delta\varphi \, d\Omega^{(in)} - \int_{\Omega^{(in)} \cup \Omega^{(ex)}} (\nabla \delta\varphi) \cdot (\nabla \lambda) \, d\Omega. \quad (9)$$

Handling the integrals in the next to last line of relation (5) in a similar way, we arrive at

$$\int_{\Omega^{(in)}} (\nabla \cdot \nabla \varphi - M_s \nabla \cdot \mathbf{m}) \, \delta\lambda \, d\Omega^{(in)} + \int_{\Omega^{(ex)}} \nabla \cdot \nabla \varphi \, \delta\lambda \, d\Omega^{(ex)} \Rightarrow$$
$$\Rightarrow \int_{\Omega^{(in)}} \nabla \cdot \nabla \varphi \, \delta\lambda \, d\Omega^{(in)} + \int_{\Omega^{(ex)}} \nabla \cdot \nabla \varphi \, \delta\lambda \, d\Omega^{(ex)} - M_s \int_{\Omega^{(in)}} (\nabla \cdot \mathbf{m}) \, \delta\lambda \, d\Omega^{(in)}$$

Using the equality (6), we transform the above expression to

$$\int_{\Omega^{(in)}} (\nabla \cdot (\delta\lambda \, \nabla \varphi) - \nabla \varphi \cdot \nabla \delta\lambda) \, d\Omega^{(in)} + \int_{\Omega^{(ex)}} (\nabla \cdot (\delta\lambda \, \nabla \varphi) - \nabla \varphi \cdot \nabla \delta\lambda) \, d\Omega^{(ex)} -$$
$$- M_s \int_{\Omega^{(in)}} (\nabla \cdot (\mathbf{m} \, \delta\lambda) - \mathbf{m} \cdot \nabla \delta\lambda) \, d\Omega^{(in)},$$

where, following the Ostrogradsky–Gauss theorem, the first terms in each of the volume integrals is reduced to the integrals over the corresponding surfaces

$$\int_{\Gamma} \mathbf{n}^{(in)} \cdot (\delta\lambda \, \nabla\varphi^{(in)}) \, d\Gamma + \int_{\Gamma \cup \Gamma_{\infty}} \mathbf{n}^{(ex)} \cdot (\delta\lambda \, \nabla\varphi^{(ex)}) \, d\Gamma -$$

$$- \int_{\Omega^{(in)} \cup \Omega^{(ex)}} \nabla\varphi \cdot \nabla\delta\lambda \, d\Omega - M_s \int_{\Gamma} \mathbf{n}^{(in)} \cdot \mathbf{m} \, \delta\lambda \, d\Gamma + M_s \int_{\Omega^{(in)}} \mathbf{m} \cdot \nabla\delta\lambda \, d\Omega^{(in)}. \quad (10)$$

In view of the relationship between the normal $\mathbf{n}^{(in)}$ and $\mathbf{n}^{(ex)}$ and the behavior of the function $\varphi$ on $\Gamma_{\infty}$, the expression (10) takes the following form:

$$\int_{\Gamma} \mathbf{n} \cdot (\delta\lambda \, \nabla\varphi^{(in)}) \, d\Gamma - \int_{\Gamma} \mathbf{n} \cdot (\delta\lambda \, \nabla\varphi^{(ex)}) \, d\Gamma -$$

$$- \int_{\Omega^{(in)} \cup \Omega^{(ex)}} \nabla\varphi \cdot \nabla\delta\lambda \, d\Omega - M_s \int_{\Gamma} \mathbf{n} \cdot \mathbf{m} \, \delta\lambda \, d\Gamma + M_s \int_{\Omega^{(in)}} \mathbf{m} \cdot \nabla\delta\lambda \, d\Omega^{(in)}$$

or, combining the integrals over the surface $\Gamma$, we get the expression

$$\int_{\Gamma} ((\nabla\varphi^{(in)} - \nabla\varphi^{(ex)} - M_s \, \mathbf{m}) \cdot \mathbf{n}) \, \delta\lambda \, d\Gamma -$$

$$- \int_{\Omega^{(in)} \cup \Omega^{(ex)}} \nabla\varphi \cdot \nabla\delta\lambda \, d\Omega + M_s \int_{\Omega^{(in)}} \mathbf{m} \cdot \nabla\delta\lambda \, d\Omega^{(in)},$$

which takes into account the continuity of $\delta\lambda^{(in)}|_{\Gamma} = \delta\lambda_{\Gamma}^{(ex)} = \delta\lambda|_{\Gamma}$. As a result, with account of condition (2), we finally obtain

$$- \int_{\Omega^{(in)} \cup \Omega^{(ex)}} \nabla\varphi \cdot \nabla\delta\lambda \, d\Omega + M_s \int_{\Omega^{(in)}} \mathbf{m} \cdot \nabla\delta\lambda \, d\Omega^{(in)} \quad (11)$$

and the variational Equation (5), in view of (9) and (11), takes the following form:

$$\delta\Psi_L = \int_{\Omega^{(in)}} \left[ \left( -\mu_0 \, M_s \, \mathbf{H}_0 + \frac{1}{2} \mu_0 \, M_s \, \nabla\varphi - 2 \, K_{anis} \, (\mathbf{m} \cdot \mathbf{p}^{\alpha}) \mathbf{p}^{\alpha} - \lambda(\mathbf{x}) M_s \, \nabla \right) \cdot \delta\mathbf{m} + \right.$$

$$+ 2 \, A_{exch} \, ((\nabla m_x) \cdot (\nabla\delta m_x) + (\nabla m_y) \cdot (\nabla\delta m_y) + (\nabla m_z) \cdot (\nabla\delta m_z)) +$$

$$\left. + 2 \, \lambda_3(\mathbf{x}) (m_x \delta m_x + m_y \delta m_y + m_z \delta m_z) \right] d\Omega^{(in)} + \int_{\Gamma} \lambda(\mathbf{x}) \mathbf{n} \cdot M_s \, \delta\mathbf{m} \, d\Gamma +$$

$$+ \int_{\Omega^{(in)}} \frac{1}{2} \mu_0 \, M_s \, \mathbf{m} \cdot \nabla \, \delta\varphi \, d\Omega^{(in)} - \int_{\Omega^{(in)} \cup \Omega^{(ex)}} (\nabla \, \delta\varphi) \cdot (\nabla\lambda) \, d\Omega -$$

$$- \int_{\Omega^{(in)} \cup \Omega^{(ex)}} \nabla\varphi \cdot \nabla\delta\lambda \, d\Omega + M_s \int_{\Omega^{(in)}} \mathbf{m} \cdot \nabla\delta\lambda \, d\Omega^{(in)} +$$

$$+ \int_{\Omega^{(in)}} (m_x^2 + m_y^2 + m_z^2 - 1) \, \delta\lambda_3(\mathbf{x}) \, d\Omega^{(in)} = 0. \quad (12)$$

Here, the first three lines are the equation at $\delta\mathbf{m}$ (three equations at $\delta m_x$, $\delta m_y$ and $\delta m_z$, each of which is equal to zero due to the arbitrariness of these variations and all others). These equations are nonlinear with respect to the sought variables, due to the first term in the third line. The integrals in the fourth line are an equation at $\delta\varphi$, which is linear with respect to the unknowns and equal to zero taking into account the aforesaid. The last two lines are two equations equal to zero at the corresponding $\delta\lambda$ and $\delta\lambda_3$, the last of which is non-linear.

The iterative methods for solving nonlinear equations essentially depend on the chosen initial approximation. If this initial approximation belongs to a neighborhood of two admissible solutions of a nonlinear equation, it is unclear to what solution the iterative procedure converges. These questions do not arise if we consider the history of the loading process (performing linearization, stepwise loading) which we are going to implement for the variational equation constructed above.

Let the variable quantities entering into (12) be represented in terms of their values at time $t_*$ (below we will denote them by "$*$") and small increments appearing as a result of transition from time $t_*$ to the nearest current time $t$: $\Delta t = t - t_*$ is a sufficiently small value. Then, we get

$$\mathbf{H}_0 = \mathbf{H}_0^* + \varepsilon\,\mathbf{h}_0, \;\; \mathbf{m} = \mathbf{m}^* + \varepsilon\,\boldsymbol{\mu}, \;\; \varphi = \varphi^* + \varepsilon\,\psi, \;\; \lambda = \lambda^* + \varepsilon\,\gamma, \;\; \lambda_3 = \lambda_3^* + \varepsilon\,\gamma_3,$$

where $\varepsilon$ is a small parameter (positive value) formalizing the concept of proximity of states at times $t$ and $t_*$. Then, bearing in mind that

$$\delta\mathbf{m} = \varepsilon\,\delta\boldsymbol{\mu}, \;\; \delta\varphi = \varepsilon\,\delta\psi, \;\; \delta\lambda = \varepsilon\,\delta\gamma, \;\; \delta\lambda_3 = \varepsilon\,\delta\gamma_3$$

and retaining in (12) only the terms of the first and second orders of smallness with respect to $\varepsilon$ (so that the variational equation will be linear in the unknowns), we obtain

- variational equation at $\boldsymbol{\mu}$:

$$\varepsilon\Big(\int_{\Omega^{(in)}}[(-\mu_0\,M_s\,\mathbf{H}_0^* + \tfrac{1}{2}\,\mu_0\,M_s\,\nabla\varphi^* - 2\,K_{anis}\,(\mathbf{m}^*\cdot\mathbf{p}^\alpha)\mathbf{p}^\alpha - \lambda^*(\mathbf{x})M_s\,\nabla)\cdot\delta\boldsymbol{\mu}+$$
$$+ 2\,A_{exch}\,((\nabla m_x^*)\cdot(\nabla\delta\mu_x) + (\nabla m_y^*)\cdot(\nabla\delta\mu_y) + (\nabla m_z^*)\cdot(\nabla\delta\mu_z))+$$
$$+ 2\,\lambda_3^*(\mathbf{x})(m_x^*\delta\mu_x + m_y^*\delta\mu_y + m_z^*\delta\mu_z)]\,d\Omega^{(in)} + \int_\Gamma \lambda^*(\mathbf{x})\mathbf{n}\cdot M_s\,\delta\boldsymbol{\mu}\,d\Gamma\Big)+$$
$$\varepsilon^2\Big(\int_{\Omega^{(in)}}[(-\mu_0\,M_s\,\mathbf{h}_0 + \tfrac{1}{2}\,\mu_0\,M_s\,\nabla\psi - 2\,K_{anis}\,(\boldsymbol{\mu}\cdot\mathbf{p}^\alpha)\mathbf{p}^\alpha - \gamma(\mathbf{x})M_s\,\nabla)\cdot\delta\boldsymbol{\mu}+$$
$$+ 2\,A_{exch}\,((\nabla\mu_x)\cdot(\nabla\delta\mu_x) + (\nabla\mu_y)\cdot(\nabla\delta\mu_y) + (\nabla\mu_z)\cdot(\nabla\delta\mu_z))+$$
$$+ 2\,\lambda_3^*(\mathbf{x})(\mu_x\delta\mu_x + \mu_y\delta\mu_y + \mu_z\delta\mu_z)]\,d\Omega^{(in)}+$$
$$+ 2\,\gamma_3^*(\mathbf{x})(m_x^*\delta\mu_x + m_y^*\delta\mu_y + m_z^*\delta\mu_z)]\,d\Omega^{(in)} + \int_\Gamma \gamma^*(\mathbf{x})\mathbf{n}\cdot M_s\,\delta\boldsymbol{\mu}\,d\Gamma\Big) = 0;$$

- variational equation at $\psi$:

$$\varepsilon\Big(\int_{\Omega^{(in)}}\tfrac{1}{2}\,\mu_0\,M_s\,\mathbf{m}^*\cdot\nabla\,\delta\psi\,d\Omega^{(in)} - \int_{\Omega^{(in)}\cup\Omega^{(ex)}}(\nabla\,\delta\psi)\cdot(\nabla\lambda^*)\,d\Omega\Big)+$$
$$+ \varepsilon^2\Big(\int_{\Omega^{(in)}}\tfrac{1}{2}\,\mu_0\,M_s\,\boldsymbol{\mu}\cdot\nabla\,\delta\psi\,d\Omega^{(in)} - \int_{\Omega^{(in)}\cup\Omega^{(ex)}}(\nabla\,\delta\psi)\cdot(\nabla\gamma)\,d\Omega\Big) = 0;$$

- variational equation at $\gamma$:

$$\varepsilon\Big(\int_{\Omega^{(in)}\cup\Omega^{(ex)}}\nabla\varphi^*\cdot\nabla\delta\gamma\,d\Omega - M_s\int_{\Omega^{(in)}}\mathbf{m}^*\cdot\nabla\delta\gamma\,d\Omega^{(in)}\Big)+$$
$$+ \varepsilon^2\Big(\int_{\Omega^{(in)}\cup\Omega^{(ex)}}\nabla\gamma\cdot\nabla\delta\gamma\,d\Omega - M_s\int_{\Omega^{(in)}}\boldsymbol{\mu}\cdot\nabla\delta\gamma\,d\Omega^{(in)}\Big) = 0;$$

- variational equation at $\gamma_3$:

$$\varepsilon\Big(\int_{\Omega^{(in)}}((m_x^*)^2 + (m_y^*)^2 + (m_z^*)^2 - 1)\,\delta\gamma_3(\mathbf{x})\,d\Omega^{(in)}\Big)+$$
$$+ \varepsilon^2\Big(\int_{\Omega^{(in)}}(m_x^*\,\mu_x + m_y^*\,\mu_y + m_z^*\,\mu_z)\,\delta\gamma_3(\mathbf{x})\,d\Omega^{(in)}\Big) = 0.$$

Here, the expressions in curly brackets with $\varepsilon$ are the variational equations corresponding to the previous step, and in the case of exact solution, they are equal to zero. Therefore, some authors exclude these terms from the following discussion. Others, on the contrary, are of the opinion that as a result of an inaccurate solution of the problem at the previous step, these terms are nonzero and must be taken into account to refine the solution at the current step. We are not aware of any theoretical estimates concerning this issue or

comparative numerical calculations made with or without consideration of these terms, and in our calculations (given below) these terms are taken into account.

The above approach to solving problems describing the behavior of magnetic materials in a magnetic field, based on minimizing the functional of the total magnetic energy, as well as the obtained variational equations are verified on the benchmark problem, which is discussed in the Appendix A.

## 5. Results of Numerical Simulation

Let us take a grain of a polycrystal or a single crystal $Ni_2MnGa$ as a ferromagnetic material. Since the characteristic dimensions of large grains are 100–200 microns, and of small grains are 1–5 microns, we choose a computational region in the form of a $L \times L$ square, where $L = 100$ nm (two-dimensional formulation) and set periodic boundary conditions

$$\varphi|_{\Gamma_1^+} = \varphi|_{\Gamma_3^-}, \quad \varphi|_{\Gamma_2^-} = \varphi|_{\Gamma_4^+};$$

$$\mathbf{m}|_{\Gamma_1^+} = \mathbf{m}|_{\Gamma_3^-}, \quad \mathbf{m}|_{\Gamma_2^-} = \mathbf{m}|_{\Gamma_4^+};$$

$$\lambda|_{\Gamma_1^+} = \lambda|_{\Gamma_3^-}, \quad \lambda|_{\Gamma_2^-} = \lambda|_{\Gamma_4^+};$$

$$\lambda_3|_{\Gamma_1^+} = \lambda_3|_{\Gamma_3^-}, \quad \lambda_3|_{\Gamma_2^-} = \lambda_3|_{\Gamma_4^+}.$$

Here,

$$\Gamma_1 : x = 0, 0 \le y \le L; \quad \Gamma_2 : y = L, 0 \le x \le L;$$

$$\Gamma_3 : x = L, 0 \le y \le L; \quad \Gamma_4 : y = 0, 0 \le x \le L.$$

The examined region measuring 100 nm × 100 nm is a unit cell, which is duplicated along the $x$ and $y$ axes, see Figure 1 (white square—unit cell). Arrows are used to represent the initial distribution of the magnetization vector (average value in the domain), such that the sample is not magnetized in the absence of an external magnetic field.

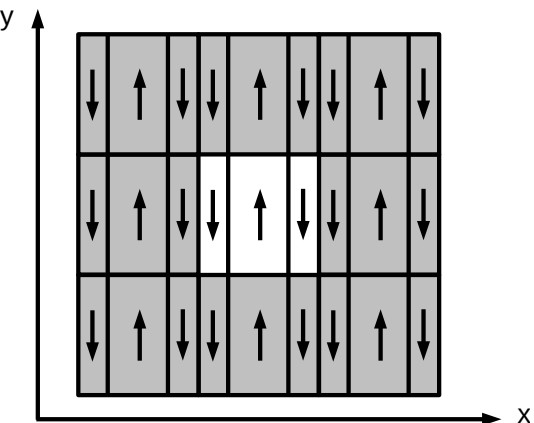

**Figure 1.** Computational domain is a white square duplicated along the $x$ and $y$ axes.

For $Ni_2MnGa$, saturation magnetization is $M_s = 6.015 \times 10^5$ A/m, magnetocrystalline anisotropy constant is $K_{anis} = 2.5 \times 10^5$ J/m$^3$ [8,16], and the exchange constant is $A_{exch} = 2 \times 10^{-11}$ J/m [8,17].

As noted in [15], the concept of domain wall thickness is a somewhat conditional one, because it is impossible to precisely determine this thickness due to a smooth change of magnetization in this area. The order of thickness is determined by the relation (see [12]) $\delta \approx \sqrt{A_{exch}/K_{anis}}$, and for the above material parameters $\delta \approx 9$ nm. The thickness by itself, $\Delta$, is determined for a 180-degree wall in two ways (see [15]): by the Lilly method $\Delta_L = \pi \delta$ and by the Landau–Lifshitz method $\Delta_{L-L} = 2\delta$, which, with account of the above value of $\delta$, gives $\Delta_L \approx 28$ nm and $\Delta_{L-L} \approx 18$ nm.

Let us introduce dimensionless quantities using the characteristic size $l_0 = 1\,\text{nm}$ (so that there are nine characteristic dimensions per one domain wall) and energy $\psi_0 = \mu_0 M_s^2 = 4.55 \times 10^5\,\text{J/m}^3$. As a result, we get the following dimensionless parameters:

$$\tilde{M}_s = 1, \quad \tilde{K}_{anis} = K_{anis}/\psi_0 = 0.54, \quad \tilde{A}_{exch} = A_{exch}/(\psi_0\, l_0^2) = 40.$$

The external magnetic field is $\tilde{\mathbf{H}}_0 = \mathbf{H}_0/M_s$.

The problem was solved for a unit 100 nm $\times$ 100 nm cell (white square in the Figure 1) by the finite element method on a regular triangular grid including 10,816 finite elements (The white square in the Figure 1 was divided into 2704 equal squares, each of which is divided by diagonals into four identical triangles with two equal sides). The sufficiency of such number of elements for convergence of the solution was determined from the numerical experiments. For the vector **m** we used a quadratic approximation, for $\varphi$ and $\lambda$— a linear approximation, and for $\lambda_3$—a constant one.

To obtain the distribution of the magnetization vector in the absence of an external magnetic field, we solved the variational equations only for **m** and $\gamma_3$, with the initial distribution of magnetization being correspondent to the distribution shown in Figure 1. In 10 steps the accuracy of the solution was $10^{-4}$.

To obtain the distribution of the magnetization vector in a magnetic field, we set at each step an increment of the external magnetic field $h_0 = 0.01$ (modulo) and realized a stepwise loading. An external magnetic field was applied along the $y$ axis, along the $x$ axis and at an angle $45°$ to the axes $x$ and $y$ in accordance with the scheme (a) $\to$ (b) $\to$ (c) $\to$ (b) $\to$ (a) $\to$ (d) $\to$ (e) $\to$ (d) $\to$ (a), where (a) is the absence of an external magnetic field, (b)—the magnetic field is $\tilde{H}_0 = 0.4$, (c)—$\tilde{H}_0 = 2$, (d)—$\tilde{H}_0 = -0.4$, (e)—$\tilde{H}_0 = -2$. Figures 2–4 show the distributions of the vector **m** in the cell, consisting of two elementary ones, with the anisotropy axis **p** directed along the $y$ axis. Red arrows show the characteristic direction of the magnetization vector in the corresponding region.

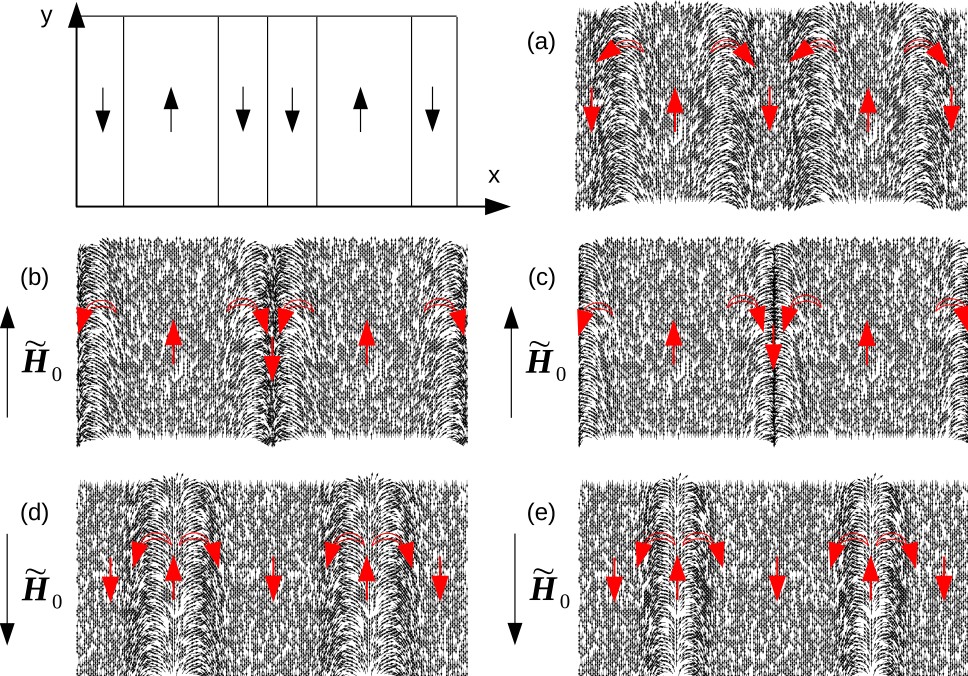

**Figure 2.** Distribution of the magnetization vector **m** in a magnetic field applied along the $y$ axis: (**a**)— $\tilde{H}_0 = 0$, (**b**)—$\tilde{H}_0 = 0.4$, (**c**)—$\tilde{H}_0 = 2$, (**d**)—$\tilde{H}_0 = -0.4$, (**e**)—$\tilde{H}_0 = -2$.

The current magnetic state of a ferromagnetic material is characterized by its specific magnetic domain structure. It varies as a result of the displacement of domain walls, the growth of some domains at the expense of others, as well as the rotation of the mag-

netization vector in the domains under the action of an external magnetic field. These changes depend on the values of the quantities determining the magnetic state of the material at the current time, as well as on their values at the previous instant of time (on the history of the process). A strong dependence on the history of the process leads to a strong magnetic hysteresis. The $Ni_2MnGa$ alloy has a weakly pronounced magnetic hysteresis loop (according to [18], the coercive field strength corresponding to this material is $H_C = 20.4$ kA/m), i.e., a weak dependence of the current magnetic state on the previous magnetic history. Therefore, in this work, we do not take it into account and do not differentiate magnetization between the points of application of the external magnetic field (a), (b), (c), and others when performing forward and backward passing through these points. In the figures given below, the positions (a), (b), (c), etc. are consistent both with the forward and backward paths of the process (coincide).

Here, in all figures in the absence of an external magnetic field one can observe the formation of the Neel walls between the domains (a different situation is hardly to be expected in the case of a two-dimensional formulation). On the average, the specimen is not magnetized. Under the applied external magnetic field, the magnetization vectors try to line up with the field, which causes the motion of the magnetic domain walls. As a result, the specimen acquires magnetization along the applied field.

In Figure 2, corresponding to the case when an external magnetic field is applied along the **p** anisotropy axis in the forward or backward direction, the two 180-degree Neel domain walls of different polarity and oriented parallel to the field meet (see positions (b) and (c)), which results in the formation of 360-degree wall, in the middle of which there is a small region of former domains with the magnetization antiparallel to the external magnetic field. This region serves as a nucleus of magnetization reversal [15] (see path (c) → (b) → (a) → (d) → (e)). As a result of magnetization reversal, the two 180-degree Neel domain walls on the path (d) → (e), showing different polarity and being parallel to the field meet again forming a 360-degree wall. They are located elsewhere and have the polarity opposite to the previous ones. In the middle of this 360-degree wall there is also a region with a magnetization antiparallel to the external magnetic field, which is now oriented in the opposite direction (compared to the previous case). This region will also serve as a nucleus of the next magnetization reversal.

As follows from Figure 2, the 180-degree domain wall has a thickness of $\sim 19 \div 22$ nm, which corresponds to the values of estimates obtained above.

In Figure 3, corresponding to the case when an external magnetic field is applied perpendicular to the anisotropy axis **p**, one can observe meeting of three Neel domain walls perpendicular to the field: 90-degree, 180-degree and again 90-degree (90-degree walls are of the same polarity, and 180-degree wall of a different polarity, see position (c)), forming, as in the previous case, a 360-degree wall. Under magnetization reversal these walls, while remaining perpendicular to the field, are displaced (arise at a different place) and their polarity is opposite to the previous ones (see position (e)). Here, the 90-degree walls are $\sim 20$ nm thick, and the 180-degree walls are $\sim 12$ nm thick.

In Figure 4, corresponding to the case when an external magnetic field is applied at an angle $45°$ to the anisotropy axis **p**, one can observe a convergence of the 135-degree and 225-degree Neel domain walls located at the same angle to the field, (see position (c)), forming, as in the previous cases, a 360-degree wall. During magnetization reversal, these walls, while remaining at an angle of $45°$ to the field, are displaced (arise at a different place) and their polarity is opposite to the previous ones (see item (e)). The thickness of the 135-degree wall is $\sim 12$ nm, and the 225-degree wall is $\sim 22$ nm.

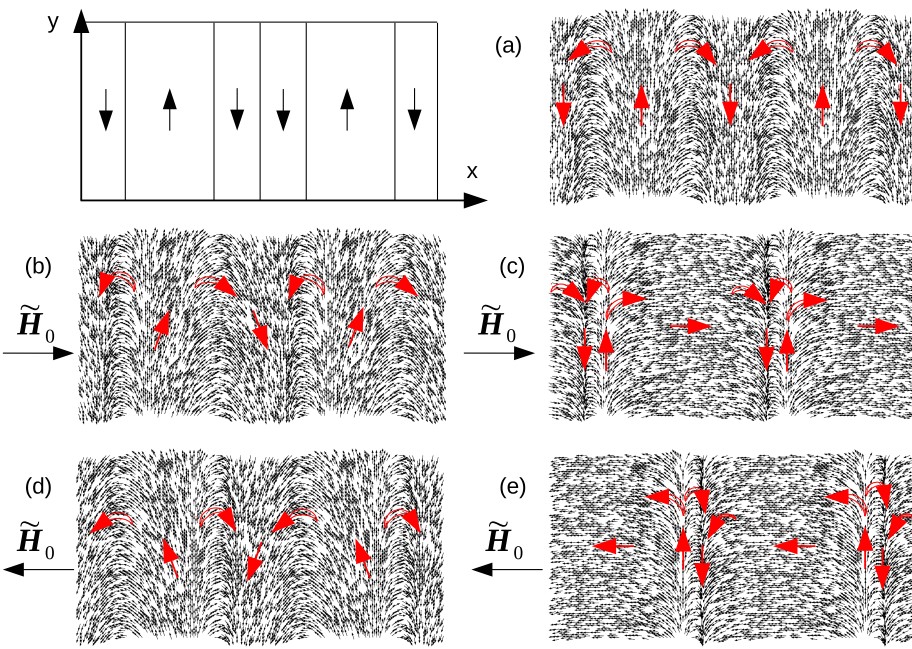

**Figure 3.** Distribution of the magnetization vector **m** in the magnetic field applied along the *x* axis: (**a**)—$\tilde{H}_0 = 0$, (**b**)—$\tilde{H}_0 = 0.4$, (**c**)—$\tilde{H}_0 = 2$, (**d**)—$\tilde{H}_0 = -0.4$, (**e**)—$\tilde{H}_0 = -2$.

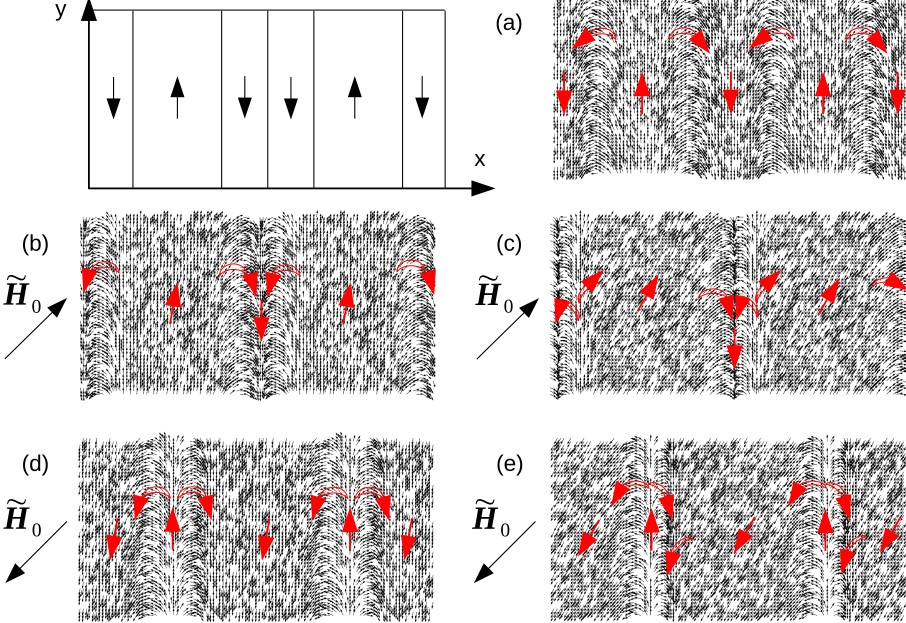

**Figure 4.** Distribution of the magnetization vector **m** in a magnetic field applied at an angle $45°$ to the *x* and *y* axes: (**a**)—$\tilde{H}_0 = 0$, (**b**)—$\tilde{H}_0 = 0.4$, (**c**)—$\tilde{H}_0 = 2$, (**d**)—$\tilde{H}_0 = -0.4$, (**e**)—$\tilde{H}_0 = -2$.

The results obtained using this approach such as the thickness of the domain wall and the formation of a 360-degree wall are in complete agreement with the known ones presented, for example, in [12,15,19–22].

As macroscopic parameters, we consider the mean value of the projection of the magnetization onto the axis along which the external magnetic field $\tilde{\mathbf{H}}_0$ is applied:

$$< m_{||} > = 1/S \int_{\Omega} m_{||} \, d\Omega,$$

and the mean value of the projection of magnetization onto the anisotropy axis **p**, which is directed along the $y$ axis:

$$< m_y >= 1/S \int_\Omega m_y \, d\Omega,$$

where $S$ is the area of the computational domain.

Figures 5 and 6 show the dependences of $< m_{||} >$ and $< m_y >$ on the modulus of the applied magnetic field $\tilde{H}_0$.

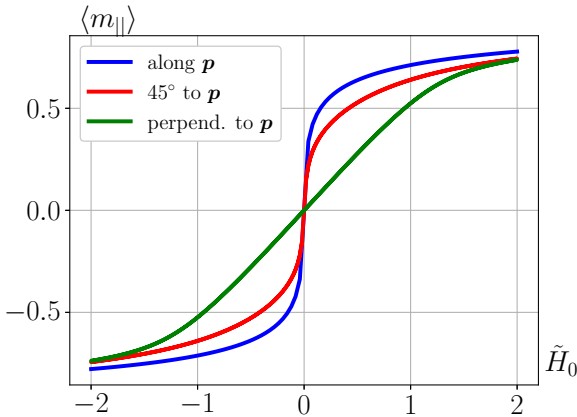

**Figure 5.** Curves of magnetization along the external magnetic field.

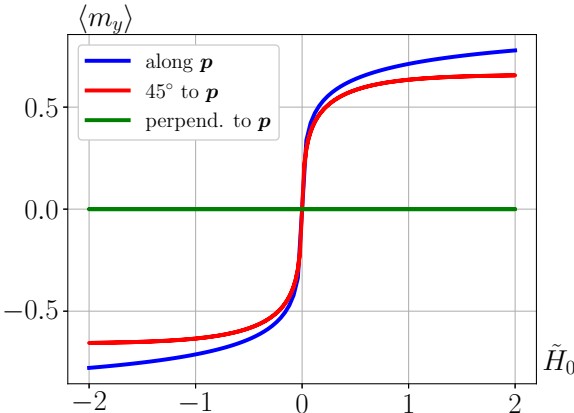

**Figure 6.** Curves of magnetization along the anisotropy axis.

The case when the external field is directed along the anisotropy axis **p** (along the $y$ axis) is denoted by the blue curve, at an angle 45° to the axis of anisotropy **p**—by the red curve and perpendicular to the anisotropy axis **p** (along the $x$ axis)—by the green curve. From these figures it is evident that the magnetic susceptibility (slope of the curve) is the greater, the smaller is the angle between the axis of anisotropy and the direction of the applied magnetic field. The hysteresis–free magnetization curve obtained for the examined material under the action of the external field directed along the anisotropy axis (blue curve), agrees well with the weakly hysteretic experimental curve given in [18].

## 6. Conclusions

In this work, a model of the behavior of a ferromagnetic material in an external magnetic field was developed within the framework of micromagnetism. Minimizing the magnetic energy functional, we constructed the nonlinear variational equations, which were numerically implemented by the finite element method using a step-by-step solution procedure. Initially, the model was applied to adjacent magnetic domains, when the

domain wall has a zero thickness. This allowed us to describe the formation of the Neel domain wall of finite thickness between the domains and construct the distribution of the magnetization vector in it in the absence of an external magnetic field. In this case the specimen is not magnetized on the average. Taking such domain structure as the initial one, we investigated its behavior (motion of domain walls and their interaction) and constructed the averaged magnetization curves under the action of an external magnetic field applied in different directions. When an external magnetic field is applied, the sample acquires magnetization along this field. In the case when an external magnetic field is applied along the anisotropy axis sequentially in the forward or backward direction, the two 180-degree Neel domain walls meet. They are oriented parallel to the applied external magnetic field, have a different polarity and forming the 360-degree wall. In the middle of this 360-degree wall there is a small region of former domains with the magnetization antiparallel to the external magnetic field. This region serves as a nucleus of magnetization reversal. In the case when an external magnetic field is applied perpendicular to the anisotropy axis, one can observe meeting of three Neel domain walls perpendicular to the field: 90-degree, 180-degree and again 90-degree. 90-degree walls are of the same polarity, and 180-degree wall is of a different polarity. As in the previous case, these walls form a 360-degree wall. In the case when an external magnetic field is applied at an angle 45° to the anisotropy axis, one can observe a convergence of the 135-degree and 225-degree Neel domain walls. These walls are located at the same angle to the field and form, as in the previous cases, a 360-degree wall. Under magnetization reversal these walls are displaced (arise at a different place) and their polarity is opposite to the previous ones. The results obtained with this approach (the thickness of the domain wall, the formation of a 360-degree wall) are in agreement with the ones available in the current literature. This and the verification performed on the standard problem confirm the adequacy of the approach developed in the article to solving problems describing the behavior of magnetic materials in a magnetic field, as well as the correctness of the obtained variational equations.

**Author Contributions:** All authors contributed equally to this work. All authors have read and agreed to the published version of the manuscript.

**Funding:** This work was fulfilled under financial support of the Russian Foundation for Basic Research through Grant 20-01-00031. The authors express their sincere thanks for this support.

**Data Availability Statement:** The data presented in this study are openly available.

**Acknowledgments:** Authors are very much obliged to L.V. Semukhina for her assistance in preparation of the paper English variant.

**Conflicts of Interest:** The authors declare no conflict of interest.

## Appendix A

Verification of the approach to solving problems describing the behavior of magnetic materials in a magnetic field, based on minimizing the functional of the total magnetic energy, as well as the obtained variational equations, is carried out on the benchmark problem muMag Standard Problem #1 (https://www.ctcms.nist.gov/~rdm/stdprob_1.html).

A rectangular plate with dimensions of $1 \times 2$ microns and a thickness of 20 nm made of permalloy is considered. The material parameters were as follows: $A_{exch} = 1.3 \times 10^{-11}$ J/m, $M_s = 8.0 \times 10^5$ A/m, $K_{anis} = 5 \times 10^2$ J/m$^3$ (for notation, see the beginning of Section 4). The easy axes of local material elements (grains) are oriented in the same direction throughout the material and nominally parallel to the long edges of the rectangle. At the initial moment of time, the sample is magnetized uniformly along its easy axis.

In this problem, the scalar potential is a continuous function over the entire computational domain containing a finite-size plate and its surrounding medium, at the boundary of which $\varphi = 0$. The magnetic field vector **H** is discontinuous in the direction normal to the plate boundary, while its tangential component remains continuous. In contrast to the field **H**, the discontinuity in **B** occurs in the direction tangential to the interface of the plate

and the surrounding medium, while the normal component remains a continuous quantity. The internal magnetic field, induced in the material, disturbs the external magnetic field, and this was taken into account when solving the problem.

In the numerical solution of the problem by the finite element method, the area occupied by the sample and a part of the external space that is 10 times larger than the sample size were considered. The 2D grid is irregular, denser in the sample and sparse in the outer space. The number of grid points in the sample is 5987 (11,672 finite elements), in the external space is 1125. Average cell dimensions is 16 nm × 16 nm. The problem is solved when the field is applied parallel to the long axis of the rectangle and when it is applied along the short axis. Figures A1 and A2 show the distribution of the remanent magnetization vectors in comparison with the results given in lu96a (https://www.ctcms.nist.gov/~rdm/std1/lu96/lu96a.html#Top). In Figure A1 an external magnetic field was applied horizontally from right to left, changes from 0 mT to 50 mT and back to 0 mT, and distribution of the remanent magnetization vectors corresponds to the last state. Here there is a quite satisfactory correspondence of our results with the results given in lu96a.

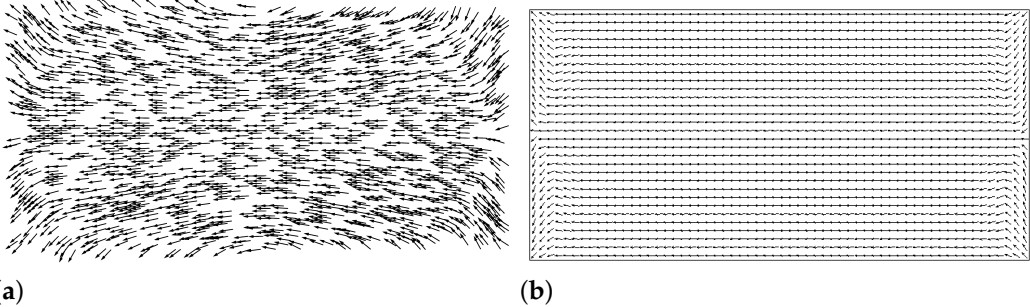

(a)                                         (b)

**Figure A1.** Distribution of the remanent magnetization vector after applying the magnetic field parallel to the long axis: (**a**) obtained by us, (**b**) lu96a.

In Figure A2 an external magnetic field was applied vertically from top to bottom, changes from 0 mT to 50 mT and back to 0 mT, and distribution of the remanent magnetization vectors corresponds to this last state.

Here, in general, there is a correspondence in the distribution of the remanent magnetization vectors, but not so satisfactory. We can explain this by the different number of grid elements in the sample (we used 11,672 while lu96a—1800). In addition, as far as we understand, lu96a does not take into account the influence of the internal magnetic field, induced in the material, on the external one.

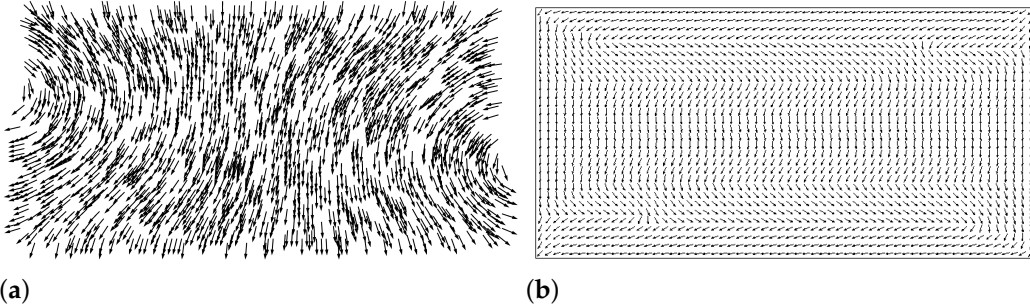

(a)                                         (b)

**Figure A2.** Distribution of the remanent magnetization vector after applying the magnetic field along the short axis: (**a**) obtained by us, (**b**) lu96a.

As noted on the site https://www.ctcms.nist.gov/~rdm/stdprob_1.html, applied above term "nominally parallel" is used to describe the orientation of fields, anisotropy axes, and sample edges. Since the domain structure can change significantly when the orientation is changed by just one degree or less, the results of micromagnetic modeling will

strongly depend on this and can only be experimentally confirmed if the actual orientation is taken into account. Today, we cannot explain the origin of this instability. This is, most probably, not a physical phenomenon, but a purely mathematical one—the instability of the procedure used for numerical implementation of the problem relative to its input parameters, and this has yet to be dealt with. The differences in the results of solving the problem under consideration are clearly shown in the table on the site https://www.ctcms.nist.gov/~rdm/mumag.org.html. This table gives some basic quantities that characterize the magnetization process, which are obtained for eight different implementations of the problem under consideration. Our results are closest to those of lu96a (https://www.ctcms.nist.gov/~rdm/std1/lu96/lu96a.html#Top), both in the description of the distribution of the remanent magnetization vectors, what was noted above, and in the values of the remanent magnetization of the entire sample after saturation along the long and short axes. The table, presented below (Table A1), shows the average values of the components of the remanent magnetization vectors, and also the values of the coercive forces given by lu96a and obtained by us. To calculate the coercive forces, the external magnetic field changed from 0 mT to +50 mT, then from +50 mT to −50 mT and returned back to 0 mT. As the table shows, the differences between our results and those given in lu96a are not so significant and can be explained by incomplete correspondence in the problem statement and different accuracy in its numerical implementation. We believe that the comparative results presented in Figures A1 and A2 and in the table fully confirm the adequacy of the approach developed in the article to solving problems describing the behavior of magnetic materials in a magnetic field, as well as the correctness of the obtained variational equations.

**Table A1.** Comparison of the obtained values.

| | Remanent Magnetization (Long Axis) | Remanent Magnetization (Short Axis) | Coercive Force (Long Axis) | Coercive Force (Short Axis) |
|---|---|---|---|---|
| lu96a | $(-0.013, 0.999, 0)$ | $(-0.987, 0.152, 0)$ | 329 Oe | 97.5 Oe |
| obtained by us | $(-0.0095, 0.933, 0)$ | $(-0.862, 0.0217, 0)$ | 307 Oe | 78 Oe |

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
