# Peer review of "The Microstructural Model of the Ferromagnetic Material Behavior in an External Magnetic Field"

_magnetochemistry, doi:10.3390/magnetochemistry7010007_

Round 1

Reviewer 1 Report

In the paper "The microstructural model of the ferromagnetic material behavior in an external magnetic field" the authors present, as far as my understanding goes, a formalism to study the micromagnetic dynamics using Lagrange multipliers to fix the constraints. In my opinion, there are some major concerns about the quality of the paper:

1) For me, the motivation of the paper is unclear. The abstract and introduction are not well organised to provide a clear logic followed by the authors. The main results and messages are not clear, in my opinion.

2) From my point of view, in the introduction the authors go over a lengthy explanation over different shape alloys which in my opinion does not contribute to a better understanding of the paper since these details are not mentioned anywhere else in the paper, such as magneto elastic effects. I believe that the paper should be reorganised and restructured to present a better and clearer message to the reader.

3) For a good evaluation of the method, I believe it would be important to check their method with the standard benchmark micromagnetic problems (for example, one can find these problems at https://www.ctcms.nist.gov/~rdm/mumag.org.html)

Reviewer 2 Report

The manuscript deals with the magnetization distribution in a ferromagnetic shape-memory system. The authors consider the magnetic subsystem neglecting the elastic degrees of freedom. Thus, they consider a simple micromagnetic problem. However, in the shape-memory systems, the magnetoelastic coupling is commonly believed to be crucial for the ordering and it provides a large complexity to the micromagnetic modeling (see e.g. refs. [10]-[12]). 

In fact, the authors study a simplified (2D) micromagnetic problem, discussing the collisions of (Neel) domain walls due to the application of an external field. The resulting magnetization structures are easy to predict being aware of behavior of the pairs of colliding Neel/Bloch domain walls in 1D (e.g. Janutka, Acta Phys Pol A 124 (2013) 23), there are also plenty of data on the simulations of the domain-wall collision for the magnetic stripes and other planar systems, thus, the mutual domain-wall annihilation or the formation of a 360deg domain wall. Hence, in the results, I do not see any novelty deserving the publication.

The authors formulate the purpose of their work at the end of the Introduction to be "the construction of a model to describe the behavior of polycrystalline material in an external magnetic field, taking into account the motion of the boundaries of magnetic domains and the rotation of the magnetic vector moment in each mesoelement, representing a grain of material". This is claimed to be the first step towards describing the shape-memory materials. In my opinion, this step has been made decades ago, however. Aren't the authors aware of the popularity of the micromagnetic packages (OOMMF, MuMax, an so on).?
Further claim is shocking to me; "The emphasis will be placed on studying the evolution of the magnetization vector through the thickness of the magnetic domain walls, which is also of interest for studying the behavior of conventional ferromagnetic materials due to the lack of sufficient researches in this area".

Reviewer 3 Report

  1. The title and the abstract of the manuscript “The Microstructural Model of the Ferromagnetic Material Behavior in an External Magnetic Field” suggest a generalized approach, whereas the manuscript begins with a specific description of the examined material, i.e. the examined Heusler alloy.

It might be useful to consider to move the specific description (lines 18-85) to some subsequent section e.g. called “Material under study”. It is enough to leave the more general part (lines 86-128) in the Introduction, in order to show the research in a context.

  1. The reference list is rather poor. The authors are right to acknowledge the works by William Fuller Brown, yet they do not mention of the Landau-Lifshitz theory (cf. e.g. G. Bertotti, I. D. Mayergoyz, C. Serpico, Nonlinear magnetization dynamics in nanosystems, Elsevier 2009), or even the reference Stoner-Wohlfarth description (it might be tempting to compare the predictions of the latter model in the context of curves presented in Figures 5 and 6). It might leave an impression to less experienced readers that the theory presented is the only possibility to perform micromagnetic simulations.

  1. The overall evaluation of the manuscript is positive. An excellent work, which provides some useful results (the authors do not forget that a model is not just a set of equations, but it requires some plausible, physics-based values for its parameters. The reviewer likes especially the attempts to assess the value of domain wall thickness with several methods).

Reviewer 4 Report

The authors present a framework for FEM in magnetic materials. They introduce the material NiMnGa. The article is very didactical and in general well written. However, it has several problems. These problems are the usual regarding articles about numerical methods.

First, the authors do not show the advantages of such formulation against other existing formulations of the FEM micromagnetics. There are already some finite elements micromagnetic packages like nmag of successors or methods or the method FEM/BEM. There is no mention to the advantages of the present formulation versus others like  the FEM/BEM approach. Also, considering the sample geometry one can simply take FD oommf software and simulate it without problems.

Therefore before publishing the authors should improve the manuscript before its publications. Moreover, the results obtained are very generic. The question is why they are discussed for a material so specific. To summarise the article can be improved.

Some remarks

-The author mentioned a russian version of the article. Is that published already in russian version? I am not sure if that is the case because it will have problems of copyright or originality. Can the authors clarify that?

-The article should discuss the advantages of such an approach vs other less complex approaches. Why do we need to take into account that? The presented geometry is suitable for a much simpler Finite Differences.

-Why is this approach more interesting for Ni2MnGa alloys? The manuscript describes that type of system but for me it is not clear why this formulation is particularly interesting for such materials. It could be used for any type of system.

-The captions of all the figures are very short. It would be better in Fig 2, 3 and 4 to include the field values in the caption. Otherwise, it is difficult to understand the difference between for example Fig.3(b) and Fig.3(c). I realize it is in the text but figures should be self explained.

-In Fig 5 and 6 it is better to include the legend or the meaning of the colors. Again, it is difficult to understand the figures having to go all the time to the text.

-The conclusion is very short. This is surprising considering that the article has almost 20 pages. A larger conclusion is necessary with the main points of the article. It starts like:”In this work, a model of the behavior of a ferromagnetic material in an external magnetic field is presented within the framework of micromagnetism. “. There are scores of micromagnetic formulations that can mention the same. So it has to be more clear about the particularities.

-Do the authors also include or are planning the elastic interactions? 

-The authors are presenting a 2D model without mentioning why a 2D model can be used, could the authors discuss that? I suppose a full 3d model will be more complicated in terms of programming but not in terms of formulation.

-In general, it would be better to include more references regarding Ni2MnGa and FEM micromagnetics. Also the mechanism with 360 degree walls is reported already in the literature, so it has to be cited properly.

-There is a section that starts like :”(weak formulation does not imply that instead of differential relations we162use equations written in the variational form …”. This is too large a discussion to be included between parentheses. I recommend a footnote or remove the parentheses and discuss the point in the text.

Round 2

Reviewer 1 Report

In this article, as far as I understood from the authors reply and modified text, the authors claim to take into account magnetoelastic effects in order to be able to describe deformable solids.
Their attempt to do so, as far as I perceived, was to solve the minimisation of the magnetic free energy with the usual constraints by using Lagrange multipliers. By solving the equations of motion including the Lagrange multipliers and with more relaxed assumptions numerically using finite element they claim, as I understood, to be able to take into account boundary and deformation effects. It is still not clear to me if this method indeed takes into account the magneto elastic effects. From my point of view, as discussed in "Theory of Magnetoelastic Effects in Ferromagnetism" Journal of Applied Physics 36, 994 (1965); https://doi.org/10.1063/1.1714293 by William Fuller Brown, it is necessary to add to the theory the displacement of the Cristal in order to observe deformation effects. Moreover, from the literature on magneto elastic effects, one constantly reads about the change in anisotropy due to deformation. This was not discussed in the manuscript, as far as I was able to understand. Even though, I believe, that a method capable to fully take into account deformations is needed, it is not clear to me if their method is indeed able to do so.

By itself, the solution of the minimisation of energy using Lagrange multipliers and finite element numerical calculations has some merit. There has been a significant effort from other groups to obtain a finite element code to solve the static and dynamical properties of the magnetization dynamics. This is not a simple task, since the finite element method introduce certain challenges that are not present for the finite difference method. For this reason, I believe that it is of extreme importance to check if indeed the proposed method can overcome these challenges and is able to get correctly some analytical predictions. Obtaining the size of the domain wall is not enough, from my point of view. I believe that, it is of utmost importance to verify the benchmark problems for validation of their results. Please compare it with the current results of the standard problems stated in https://www.ctcms.nist.gov/~rdm/mumag.org.html

Reviewer 2 Report

Regarding the two points at the beginning of the Author's Response, I don't think they are the opinions formulated by me, (perhaps, they are based on the quiz, while, they are rather unfortunate). In terms of the style, I think, there is a lot of unnecessary informations regarding the history of the micromagnetism and details of the shape-memory magnets which cover the merit of the work, (however, I don't have objections to English language), In terms of the applicability of the description, whole my critique has been expressed in the comments of the report.

In terms of the merit, the explanation of the purpose of the work by the authors has convinced me. They perform full micromagnetic modeling of a ferromagnetic medium looking for the conditional extremum of the action functional instead of solving integro-differential LLG equation. This approach is not expected to add any value to our knowledge on the magnetic phenomena itself (it is not optimized since a complexity is due to the increased number of the freedom degrees), however, it allows one for composing the magnetization dynamics with nonmagnetic freedom degrees within an universal (Lagrange) formalism and for performing numerical simulations with use of universal finite-element platforms.

At present, I find the work to be very interesting and to deserve of being communicated to the audience. Although, I regret that none test of the numerical efficiency of the proposed method of solving the micromagnetic problems is provided. Anyway, I wolud suggest shortening the Introduction, which should proportional to the main body of the text and focused of the problem that is realy solved. At present, it discourages the reader (at least the reviewer).

Reviewer 4 Report

The authors have replied my questions. They have added more material that makes the content more complete.

Author Response

The authors have replied my questions. They have added more material that makes the content more complete.

English language and style are fine/minor spell check required.

Response. We appreciate the reviewer's agreement with our responses. We checked the spelling again and made the necessary corrections to the words where we were able to detect its violation. In the revised once more version of the article, we added, at the urgent request of two other reviewers, positive results of testing the approach proposed in the article on standard micromagnetic problem (see Appendix).

Remark. All changes and additions made to the new version of the article are shown in italics.